# Compression of Text in Selected Languages—Efficiency, Volume, and Time Comparison

**DOI:** 10.3390/s22176393

**Published:** 2022-08-25

**Authors:** Beniamin Stecuła, Kinga Stecuła, Adrian Kapczyński

**Affiliations:** 1Faculty of Applied Mathematics, Silesian University of Technology, Akademicka 2A, 44-100 Gliwice, Poland; 2Faculty of Organization and Management, Silesian University of Technology, Akademicka 2A, 44-100 Gliwice, Poland

**Keywords:** processing, compression, coding, compression algorithms, languages

## Abstract

The goal of the research was to study the possibility of using the planned language Esperanto for text compression, and to compare the results of the text compression in Esperanto with the compression in natural languages, represented by Polish and English. The authors performed text compression in the created program in Python using four compression algorithms: zlib, lzma, bz2, and zl4 in four versions of the text: in Polish, English, Esperanto, and Esperanto in x notation (without characters outside ASCII encoding). After creating the compression program, and compressing the proper texts, authors conducted an analysis on the comparison of compression time and the volume of the text before and after compression. The results of the study confirmed the hypothesis, based on which the planned language, Esperanto, gives better text compression results than the natural languages represented by Polish and English. The confirmation by scientific methods that Esperanto is more optimal for text compression is the scientific added value of the paper.

## 1. Introduction

Data compression is one of the achievements of the information revolution. This revolution has changed our lives, especially in the field of application and the use of digital content. Despite the fact that the phenomenon of data compression is often not visible, it has become more and more ubiquitous. Data compression is part of the information technology of our lives. It is one of the enabling technologies for each of the aspects of the multimedia revolution. Images, audio, and video files are available for placement on the website thanks to compression algorithms. For a long time, data compression was the domain of relatively small groups of engineers and scientists, but now it is widespread [1]. The daily activity of people on the Internet is possible thanks to compression. Nowadays, more and more information is in digital form. The number of bytes needed to represent multimedia data can be huge. Although hardware manufacturers and companies produce tons of hardware in an attempt to provide a better solution to work with large amounts of data, it is almost impossible to keep those data uncompressed [2].

The explosive growth in data generation led to development in data transmission and storage. The possibilities and efficiency of transmitting and storing content depend, on the one hand, on technology, and on the other hand on input data. Nowadays, more and more data are generated, and they must be stored. Ways to optimally store data should be looked for. This refers to different types of files, including text files. Text data can be compressed and stored in different languages, and a lot of them use English; for example [3,4,5,6,7]. However, it turns out that the issue of which language is the most optimal in terms of compression efficiency, time and volume remains insufficiently researched. There is no research in the literature or practice on the problem of which language is most optimal for compressing text files. There is also little research on artificial languages and their application in computer science, especially in text compression. Studies on the compression and storage of text do not take into account the influence of the type of language. The authors of this article noticed that the length (volume) of the same text for different languages is different. It means that as input data for compression, the same text differs in the case of different languages. Therefore, the authors decided to conduct research in the field of comparing the compression of the same texts for selected languages. The authors studied the Esperanto language, which is an artificial (planned) language, in the context of its role in compression, and decided to study the problem of text compression depending on the language. The authors decided to investigate effects that would be obtained by compressing a text in Esperanto. The Esperanto language is characterized by simplicity, regularity, and often repetitive fragments [8]. The words in this language are composed of unchanging morphemes, which makes it naturally suitable for computer compression. In order to compare the effects of compression, the authors also chose Polish and English for the research. The Polish language was chosen due to the fact that it is the native language of the authors of the research. In turn, the English language is one of the most popular and widely used languages; moreover, it is the language in which this publication is written.

A new and thus-far unexplored direction of research is searching for an optimal language suitable for text compression. The authors undertook the study, choosing Esperanto due to the above-mentioned features of this language. The main goal of this research was to study the possibility of using the Esperanto language for text compression and to compare the results of text compression in Esperanto, Polish, and English. The authors formulated the research gap as the lack of direct text comparison of compression in the planned language Esperanto and natural languages. In the research, the following research question was asked: “Is the planned language Esperanto more suitable for the compression process than the natural languages represented by Polish and English?”. The hypothesis was formulated as: “Planned language Esperanto gives better text compression results than natural languages represented by Polish and English”. The scientific added value of the research was determining the language that gives better text compression results. The scope of the research included the following:Study of Esperanto grammar and vocabulary.Development of the theoretical background of the solution.Selection of research tools.Finding the text in Esperanto, Polish, and English as input data for the compression process.Implementation of a program for text compression.Process of text compression.Analysis of the results.Development and update of the program in the future.

The paper is divided into five sections. Section 2 discusses related work on the topic of text compression. Section 3 describes the materials and methods used in the research. This section includes the selection of the planned language Esperanto, input data, a research tool, and a developed program for text compression. The compression program was developed by the authors. Section 4 presents the results of text compression and an analysis of the results of this compression. Section 5 includes conclusions.

## 2. Related Work

Data compression is a process that reduces the size of the data, removing excessive information [9]. A smaller data size is suitable because it reduces costs. Compression aims to reduce redundancy in stored or communicated data, thus increasing the effective density of the data. Compression is the representation of data in a reduced form so that data can be saved by using a small amount of storage and sent with limited bandwidth. In practice, data compression is based on reducing the size of the text by finding regularities such as long sequences of the same characters or repetitive sequences of letters and words [1]. Data compression can be divided into two techniques, which are the following: lossless and lossy [1,10]. Lossless compression reproduces the data perfectly from its encoded bit stream. The original information content can be recreated in its original and unchanged form. This type of compression is only about changing the recording of the information. In lossy compression, in turn, less significant information is removed. It is a more efficient type of compression. It keeps the most important properties of the output information, but it may lose some details. It is often used to compress graphics and video files in such a way that only information noncrucial to the user is lost. An example is playing a movie with a lower resolution than the original. Computer compression usually needs to be uncompressed to be readable for humans [1]. There are many types of lossless text compression. They include the Burrows–Wheeler transform, Huffman coding, arithmetic coding, run-length coding, Deflate, Lempel–Ziv 77 (LZ77), Lempel–Ziv–Welch (LZW), GNU zip (Gzip), Bzip2, Brotli, and many more [1,11]. Some statistical methods assign a shorter binary code of variable length to the most frequently repeated characters, and examples of this method are Huffman and arithmetic coding. According to [12], the Huffman coding is one of the best algorithms in this category. As [13] claims, Deflate provides slightly poor compression, but its encoding and decoding speeds are fast. In addition, in the literature, a lot of research on the compression issue can be found; for example [14,15,16,17].

Compression is a subject of some research that is described in the literature. The authors of [18] proposed a novel frequent pattern-mining-based Huffman encoding algorithm for text data and used a hash table in the process of frequent pattern counting. Their algorithm operates on pruned set of frequent patterns, and is efficient in terms of database scanning and storage space, reducing the code table size. The main objective of the paper [19] was to identify which compression technique was better in text, image, and audio compression applications. According to authors, for text files, due to a shorter compression ratio and lower compression time, the Huffman Algorithm is recommended. For the image files, Lempel–Ziv was better compared to the Huffman Algorithm because of the huge difference between their compression ratios and compression times. For audio files, the comparison between the compression times of the two algorithms was inconclusive because there was no significant difference between the two. Another research study [20] focused on an engineering perspective of Data Mining, using it as a tool for efficient data compression. The research exploited the principle of assigning shorter codes to frequently occurring patterns in relation to the single-character-based code assignment approach of Huffman encoding. Research in [4] presented a new algorithm for the compression of very short text messages. Their algorithm converts the input text consisting of letters, numbers, spaces, and punctuation marks commonly used in English writings to a format which can be compressed in the second phase, which is a transformation reducing the size of the message by a fixed fraction of its original size.

The topic of compression is also related to the Internet of Things (IoT) issue. As [21] claims, a key challenge of the modern digital world is reducing the size of the transmitted data without sacrificing their quality. A natural solution is to compress data at the sensing devices. Therefore, this problem is the subject of many studies. The authors of [22] proposed a novel method to significantly enhance transformation-based compression standards, such as JPEG, by transmitting much fewer data from an image at the sender’s end. They proposed a two-step method by combining the state-of-the-art signal processing-based recovery method with a deep residual learning model to recover the original data. The authors of [21] introduced Sprintz, which is a compression algorithm for multivariate integer time series that achieves state-of-the-art compression ratios across a large number of publicly available datasets. The work of [23] describes a novel multivariate data compression scheme for smart metering IoT. The proposed algorithm exploits the cross-correlation between different variables sensed by smart meters to reduce the dimension of the data. The studies of [24] presents an optimal compression technique using CNNs for remote sensing images. The method uses CNN to learn the compact representation of the original image which held the structural data and was then coded by the Lempel–Ziv-Markov chain algorithm. There is also more work on lossless compression algorithms suitable for IoT; for example, [25,26,27,28].

In the literature, there are more studies that focus on the compression of English texts; they include, for example: [29,30,31,32,33,34]. There are not many works that focus on different languages. The first example is [35], which shows transliteration-based Bengali text compression. Other studies focused on German [36,37] and Arabic [38,39], but there is also a study on compression comparison of English, German, French, Japanese, and Chinese [40], and English, German, French, Italian, Czech, Hungarian, Finnish, and Croatian [41]. When it comes to Polish, there are not many studies [42,43,44,45]. However, based on the literature review, it can be claimed that there are no studies on compression comparison of English, Polish, and Esperanto at all—and this justifies the authors’ need to conduct research in this field.

## 3. Materials and Methods

### 3.1. Selection of the Planned Language Esperanto

In 1887, Ludwik Zamenhof officially published the first version of the international language he had created, known as Esperanto. This language was created in order to reconcile nations in the territory he inhabited. These nations communicated in three or more languages, which resulted in frequent misunderstandings, divisions, and prejudices [46,47]. Due to his language, Ludwik was nominated eight times for the Nobel Peace Prize [48]. In 1908, the Universala Esperanto Asocio (UEA), i.e., the World Esperantists’ Union, respected by such organizations as the United Nations, UNICEF, UNESCO, the Council of Europe, the International Organization for Standardization (ISO), and the Organization of American States, was funded [49].

Esperanto is an artificial language. Its main characteristic is its regularity (due to the immutability of morphemes). Due to its unambiguous and analytical nature, Esperanto is very precise. It should be noted that parts of speech are easily distinguished due to specific grammatical endings. In the context of sentence order, the order of words is characterized by great freedom. The grammar of the language is described in Zamenhof’s Foundation of Esperanto (epo. Fundamento de Esperanto) [8].

The vocabulary of Esperanto has its roots in European languages, mainly Romance (around 75%) and Germanic (around 20%) [50]. There are several hundred words in the Esperanto Foundation, and nowadays, their number exceeds a thousand. All of these words form a complete language, as they also form the basis for more complex expressions that are formed by putting together existing words.

Thus, the vocabulary in Esperanto forms a certain system that allows us to obtain a range of meaning many times greater than in the case of languages without extensive prefixing and suffixing. Therefore, it should be stated that within the framework of the currently existing database, it is possible to create concepts that do not yet have equivalents in natural languages.

Esperanto was not the first artificial (planned) language. It is worth mentioning the Volapük language, which directly preceded Esperanto [51]. The creator of this language, Johann Schleyer, did not agree with any reforms proposed by the academy of this language. The creator’s resistance led to a schism, and then to a decline in the language’s popularity. Volapük was a regular and schematic language, but the extensive and difficult-to-remember and pronounce vocabulary meant that even the creator himself was not able to use it fluently [52]. Nowadays, the language is still used on the Internet [53]. However, it appears only in written form—in oral form, it is used only by hobbyists. After Esperanto, many other artificial languages have emerged, such as Occidental (Edgar von Wahl, 1922), Novial (Jesperen, 1928), Interlingua (1951), and Romanid (Zoltan Magyar, 1956) [54]. Moreover, some languages were directly based on Esperanto, including Ido (1907), Reform Esperanto (1910), and Latin Esperanto (1911) [54]. However, it should be emphasized that all these popular languages had the disadvantage of being “naturalistic”. This means that they imitated natural languages to such an extent that they consequently lost their original feature; that is, regularity.

Based on the idea accompanying the Esperanto language, a movement called “Esperantism” was founded. Its main assumptions were the following [8]: Striving to introduce a neutral human language around the world;Popularization of Esperanto through practice, including increasing the library of sources (both original and translations);Waiver of rights by the author of the Esperanto language—Esperanto belongs to everyone;No one can introduce new rules into the language. The only source is the Esperanto Fundament;Every user of Esperanto is an Esperantist.

It is worth noting that these goals are very similar to the currently popular idea behind many programming languages or software, which is creating and sharing open source code. Due to the simplicity, regularity, and specific grammar and vocabulary of the Esperanto language, the authors decided to undertake research on the use of this language in computer science. The authors studied the grammar, vocabulary, and principles of the language. Based on the analysis, the authors concluded that Esperanto is optimal for use as a human language, which will be easily analyzed and processed by computer software. It is also worth noting that, thanks to the popularity of this language, numerous source materials that can be used as valuable research sources were created.

Moreover, it is worth noting that due to its regularity, the Esperanto language was applied in many projects and became the subject of many studies in the field of computer science; for example, creating an algorithm for morphological segmentation of Esperanto words [55], a study on Zipf’s law [56], building a model of the world context [57], enhancing the development of human–machine communication [58]. Moreover, the authors [59] focused their paper on discussing the digital presence of Esperanto. They assessed its digital vitality on the basis of its language ideology and other sociolinguistic data. Other researchers [60] quantified the irregularity of different European languages belonging to four linguistic families and an artificial language (Esperanto). They worked on modifying a well-known method to calculate the approximate and sample entropy of written texts. They based their method on the search for regularities in a sequence of symbols and consistently distinguishing between natural and synthetic randomized texts. The mentioned reviewed research examples, followed by the authors’ experience, inspired the authors of this article to conduct their research on text compression in Esperanto.

### 3.2. Input Data

For a text compression program to produce meaningful results, it had to operate with varied and long text as input research material. The authors had to look for a text available in each of the three selected languages. Therefore, the study uses the entirety of the novel by the Polish writer Henryk Sienkiewicz—*Quo vadis*. The author of the novel was the first Polish writer to receive the Nobel Prize for Literature—this was in 1905 [61]. *Quo vadis*, written in Polish and published in 1896, was recognized as a worldwide bestseller at that time [62]. The novel has been translated into over 30 languages. It is a novel in the public domain with readily available translations, including English and Esperanto. The selection of this novel as the subject of the research carried out allowed us to obtain input text (input data) with very similar characteristics and length, minimizing the differences that could affect the result of the experiment in an unexpected way.

The authors searched the database of novels in the public domain and found the novel *Quo vadis* by Henryk Sienkiewicz. The sources of the downloaded novel and its translations were the following:In Polish—wolnelektury.pl (accessed on 5 February 2022) [63];In English—gutenberg.org (accessed on 5 February 2022) [64];In Esperanto—tekstaro.com (accessed on 15 January 2022) [65].

The data were saved in the .txt format in 8-bit Unicode Transformation Format (UTF-8) coding.

The most popular compression algorithms were used to compress the text. Four algorithms selected for the research include the following:Zeta Library (zlib);Lempel–Ziv–Markov chain algorithm (lzma);Bz2;Lz4.

The processing and presentation of the collected data included the following:Examination of the compressed text and measurement of the compression time;Data presentation in text form was provided through a specially created class, which stores the collected data and writes them to the console;Entering data into the Microsoft Excel spreadsheet table and presenting the results in the form of tables and graphs.

### 3.3. Research Tools

The programming language used in the compression program in this study was Python 3.9.5, a general-purpose high-level language. Its author is Guido van Rossum [66]. The greatest advantages of Python include the readability and transparency of the code, which facilitate code management, development, and debugging. Thanks to these advantages, working in this language is efficient, and the programmer can pay more attention to the logic of the implemented algorithm. Thanks to the interpretability, errors can be easily spotted—in the event of a stoppage, the program shows the place that requires attention. Python is a dynamically typed language, so there is no need to specify the type of variable to be used; it is possible to use the variable without having defined it previously [67]. The Python programming language is a valued language in the world, which is confirmed by the fact that it was chosen by Alphabet Inc., Mountain View, CA, USA (a conglomerate holding company created through a restructuring of Google) as one of its main programming languages [67].

PyCharm is an Integrated Development Environment (IDE) for professional Python developers. Its producer is the software producer Jet-Brains, known from other projects, such as IntelliJ IDEA, RubyMine, and the Kotlin language [68]. It was chosen as a research tool in this study due to its comprehensive approach to programming, which includes automated code refactoring, autocompletion, convenient keyboard shortcuts, and suggestions for improving readability based on the PEP8 standard.

The UTF-8 format was selected as a text form. It is a Unicode encoding system, created in accordance with the American Standard Code for Information Interchange (ASCII) [69]. UTF-8 is used by over 98% of the largest websites in the world and by over 97% of all types of websites [70]. It was the optimal choice for research due to its compatibility with the Polish, English, and Esperanto alphabets.

### 3.4. Developed Program for Text Compression

The developed text compression program initially loads the input data in each of the languages that are the subject of research. After loading the data, the developed program compresses and saves the work. Finally, the results obtained are presented. Figure 1 presents pseudo code for the developed program. The program reads the text of the book in binary form and stores it in the *text* variable. Each language has a separate text file named after the language/notation abbreviation adopted by the authors. Their names are the following:pl—stands for Polish language;en—stands for English language;eo—stands for Esperanto language;eox—stands for Esperanto language in notation x.

**Figure 1 sensors-22-06393-f001:**
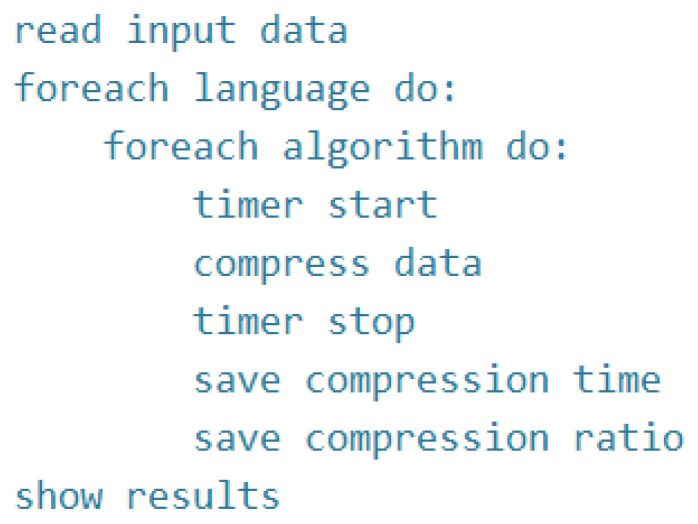
Pseudo code for the developed program.

Each language has its own unique features presented in Figure 2 and described below, based on the study and the experience of the authors. 

Text in Polish—the Polish language is characterized by having characters in its alphabet that are not in ASCII encoding, so each such letter takes up more space.

Text in English—the English language is characterized by the fact that the basic words are short, but the names of more complex concepts often require the use of a few words, so that the volume of the translated text is larger than the original.

Text in Esperanto—Esperanto is characterized by longer base words than the English language, but shorter words for more complicated concepts. Therefore, long and complicated texts are smaller than those in Polish and English.

Text in Esperanto in x notation—in order to exclude the influence of encoding on compression, text written with x notation was used. 

Letters with signs above them (for example “ŝ” or “ŭ”) have their ASCII equivalents followed by the letter x (e.g., instead of “ŝaŭmo” it would be “sxauxmo”). These letters appear in a minority of words, so most of the text remained unchanged. Table 1 shows the data on uncompressed text. It can be noticed that the text length is the same for the eox and eo versions. Encoding letters with special signs takes two bytes, and converting them to two ASCII characters results in the encoding of two single-byte letters, which ultimately does not affect the size of the text.

Table 2 shows the sample text from *Quo vadis* in different languages and the adequate number of characters.

The data listed in the console included, inter alia, the following:The length of the uncompressed text;The name of the language for which the compression is performed;The name of the compression algorithm;The compression time;The percentage of space taken in relation to uncompressed text;Bytes space taken up by compressed text.

Table 3 shows the data contained in the console.

The timeit library was used to measure the compression time. It allows an easy measurement of the time of repeatedly providing a function. It gives a value that is the time in seconds of all the completed loops, and the value is finally divided by the number of loops to obtain the unit information. In the research, there were five hundred repetitions of each measurement.

During the research, the authors improved the text compression program. Examples of changes are the following:Adding a text version with x notation;Collecting the length of compressed texts in bytes;Collecting the length of uncompressed texts in bytes;Adding more algorithms in order to verify the set hypothesis and obtain more reliable results.

## 4. Results

### 4.1. Text Volume before and after Compression

The subject of compression was Henryk Sienkiewicz’s novel *Quo vadis*. Figure 3 shows the uncompressed text volumes in each language. The unit is bytes. It is worth noting that the uncompressed Esperanto text occupies less space than the English and Polish texts. Compressed text, on the other hand, allows us to save even more space, as can be seen in the graphs in Figure 4.

Subsequently, the authors compared the texts after compression in terms of volume. Table 4 shows the collected data on volumetric compression efficiency. Data in rows correspond to individual algorithms; columns—languages/notations; and the values indicate the percentage of volume of the compressed text in relation to the uncompressed text-Equation (1), that is, written in an open, human-readable form, in a .txt file in UTF-8 encoding. The applied algorithms are deterministic, which means that their multiple provisions gave the same results each time.
(1)$$E_{c}=\frac{c_{a}\;}{c_{b}\;}\times 100\%$$
where: $E_{c\;}$—efficiency of compression; $c_{a}$—text volume after compression; $c_{b}$—text volume before compression. 

Figure 5 shows the compression efficiency in the form of graphs. It can be concluded that, in terms of efficiency, compression is the least affected by Polish, then English, and Esperanto is the best. It is worth noting that the very application of the x notation, despite the fact that it does not change the text volume, improves compression. It is minimal: below one tenth of a percentage point; however, it still is an improvement. The obtained results confirm the hypothesis presented in the research, which was formulated as: “Planned language Esperanto gives better text compression results than natural languages represented by Polish and English”.

The authors decided to make comparisons of text compression in each of the analyzed languages, adopting a common denominator in these comparisons. In other words, the authors decided to refer each of the comparisons to the original text of *Quo vadis* in Polish. They referred compressed texts in individual languages to the uncompressed original text, i.e., in Polish. The equation to determine the effectiveness in relation to the volume of the text in Polish is presented as Equation (2). The results of the analysis are presented in Table 5 and in the graph in Figure 6. Based on this comparison, it was possible to visualize the practical percentage savings, which in the case of the LZ4 algorithm was not 1.35 pp, but 2.18 pp.
(2)$$E_{c_{pl}}=\frac{c_{a}\;}{c_{b_{pl}}\;}\times 100\%$$
where: $E_{c_{pl}}$—efficiency of compression in relation to text volume in Polish; $c_{a}$—text volume after compression; $c_{b_{pl}}$—text volume in Polish before compression.

### 4.2. Time of Compression

Table 6 presents the collected data on the time of a single compression for the selected algorithm and language/notation. These data are presented graphically in Figure 7. Based on these charts, it can be observed that there are no major differences in the execution time of individual algorithms in different languages. The data shown were performed with five hundred repetitions of each measurement. Then, these measurements were summed and divided by their number to give the average. The total compression time, including data in all languages, all algorithms, and five hundred repetitions for each of them, was over 23 min. When analyzing the compression data, one can only conclude that there are no clear differences in compression times between languages. It is worth noting that lzma algorithms take the longest time to compress. This is in accordance with the results of another study [13], which confirm that this algorithm has the lowest compression speed compared to other algorithms.

### 4.3. Additional Comparison of Compression of Text Translated in Google Translate

To support the received results, the authors decided to carry out an experiment. Using the document translation option in Google Translate (GT), the entire novel of *Quo vadis* was also translated from English (as it is the default language for GT) into eight other languages. Then, the compression was performed with the developed program and the results were presented in Figure 8 and Figure 9. As can be observed, among the tested languages, Esperanto achieved the best results in terms of saving disk space. It is also worth noting that the compression efficiency depends only on the language of the text and is almost independent of the algorithm used for compression (Esperanto in each algorithm got the best results, and Russian the worst).

## 5. Conclusions

The authors developed a program for text comparison that can effectively compress text with the use of various algorithms, perform an examination of the work time and rate of text compression, as well as present the collected information. The created program allows us to compare any texts based on various compression algorithms in terms of the time needed to perform the work and the final compression effect. It is able to operate on texts in any language and various recording formats, as it operates on binary notation. The program uses four compression algorithms, and all of them indicated that the planned language Esperanto in any form of writing allows for a higher degree of compression than the studied natural languages. Therefore, Esperanto has been concluded to provide a more space-saving way to write information for storage on a computer. This conclusion is significant, because nowadays more and more data are produced, and they must be stored. As the literature review revealed, methods for optimal data storage are being searched for. Esperanto gives better compression results because this language is more precise, unequivocal, and regular. It gives a chance for space savings, which in terms of the Internet of Things is very important. Based on the results, it can be claimed that Esperanto gives us a chance to substantially save resources. This conclusion creates a value-added and opens a broader perspective for further research on the use of Esperanto in data compression.

The developed text compression program was used successfully to conduct an analysis of the efficiency of text compression. It correctly compresses and collects data on compression rate and time. In the study, the main goal, which was to study the possibility of using the Esperanto language for text compression and to compare the results of text compression in Esperanto, Polish, and English, was realized. The authors found the answer to the set research question, which was as follows: “Is the planned language Esperanto more suitable for the compression process than the natural languages represented by Polish and English?”. The results enabled a positive answer to this question. In the criterion of the efficiency of text compression, the results confirmed that Esperanto receives better results. The obtained results confirmed the hypothesis—“Planned language Esperanto gives better text compression results than natural languages represented by Polish and English”. The experiment showed a higher compression ratio for Esperanto language compared to the natural languages—Polish and English. It took up more than two percentage points less space than in the case of the compression of the English text. The satisfactory result of the research was obtained due to the fact that the Esperanto language is based on unchanging morphemes, as well as on regularity combined with the absence of exceptions.

The program for text compression has potential to be developed in future in the several ways. First of all, the research can be improved by adding more natural languages to the comparison. Another way is to increase the base of various compression algorithms used in the study. After receiving the best results on comparison text in Esperanto, it can be claimed that it is worth adding and studying more text sources and comparing the results. Similarly, it is worth analyzing longer and shorter texts to see if the compression in Esperanto would give better results than in this study. Moreover, a big potential for study gives replacing the compression of the entire text with the use of algorithms using Machine Learning (ML) and Neural Networks (NN). It is also important to conduct research on saving space, especially in the context of the Internet of Things and increasingly used smart sensors. Therefore, an interesting solution would be to use Esperanto in applications such as speech-to-text and text-to-speech language rather than what is usually used: English. Broader research on the use of Esperanto to store larger texts is absolutely justified, and therefore, the authors plan to deal with them in the future.

## Figures and Tables

**Figure 2 sensors-22-06393-f002:**
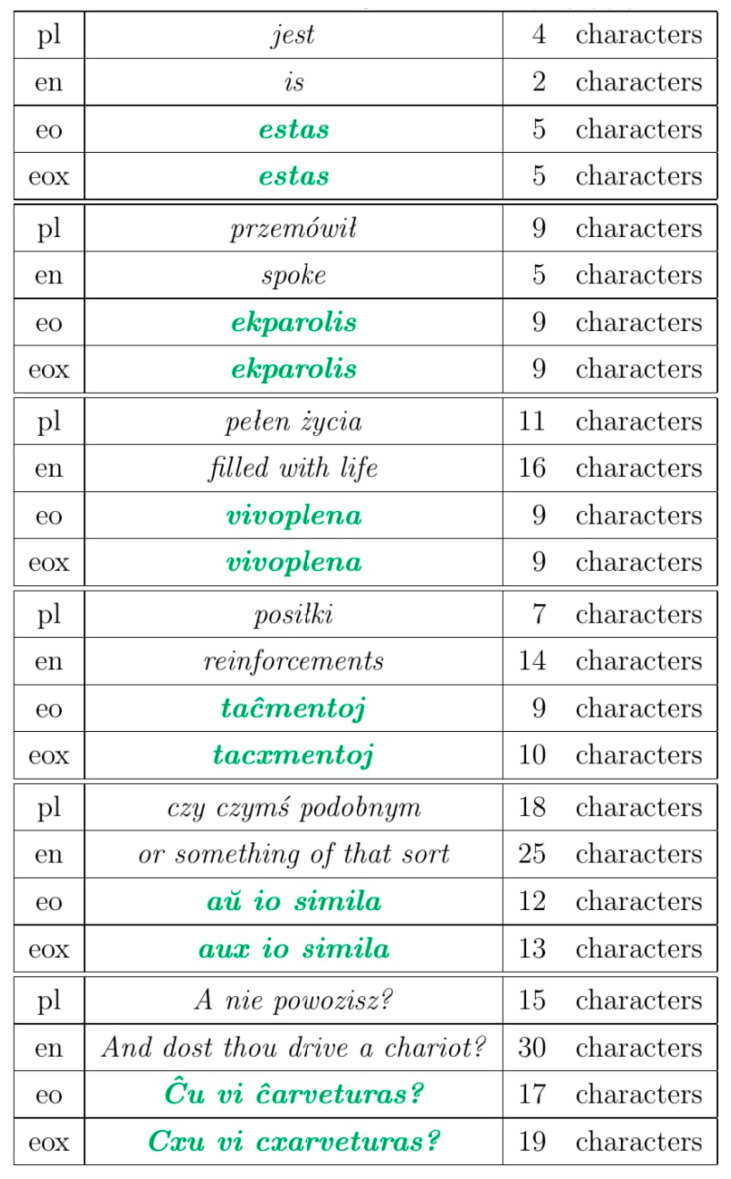
The comparison of the lengths of the characters in each language.

**Figure 3 sensors-22-06393-f003:**
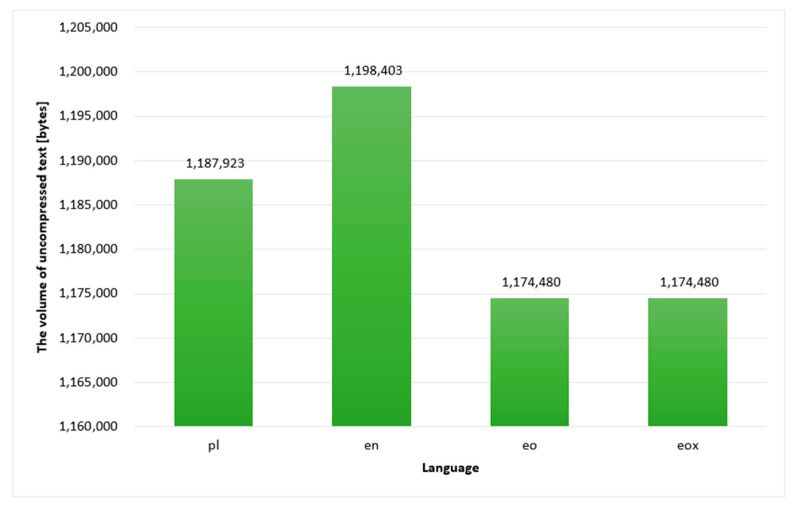
Volume of uncompressed text in the given languages.

**Figure 4 sensors-22-06393-f004:**
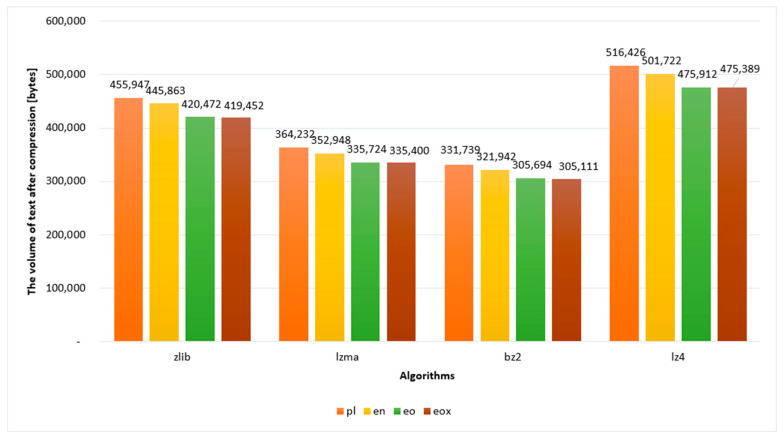
Volume of compressed text in the given languages.

**Figure 5 sensors-22-06393-f005:**
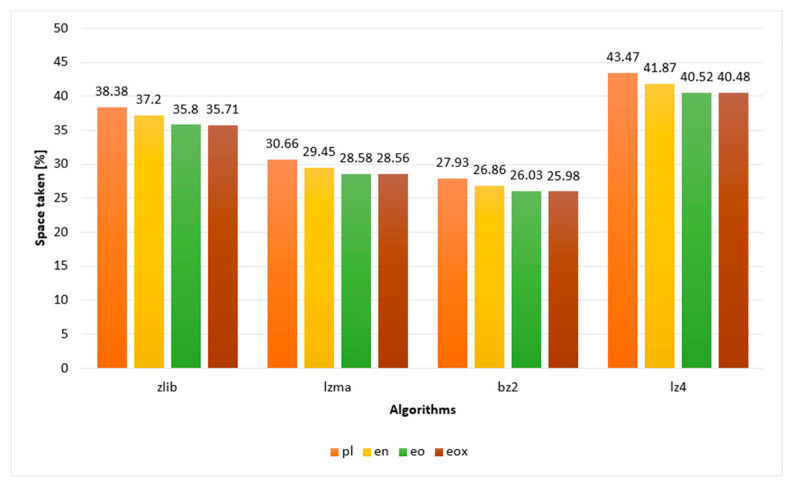
The efficiency of text compression.

**Figure 6 sensors-22-06393-f006:**
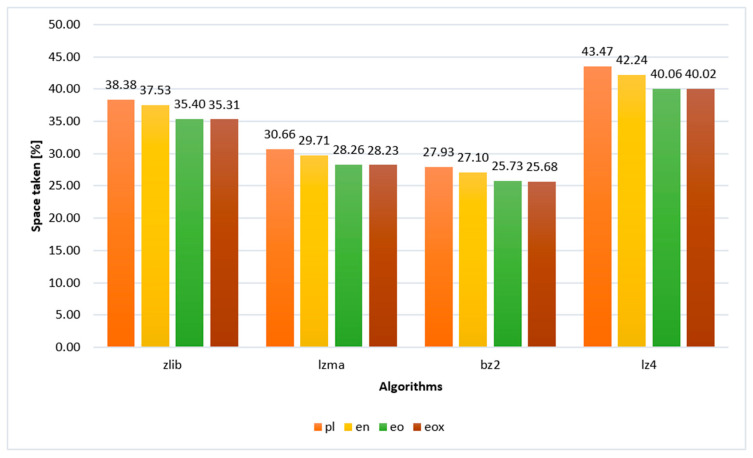
The efficiency of text compression in relation to text volume in Polish.

**Figure 7 sensors-22-06393-f007:**
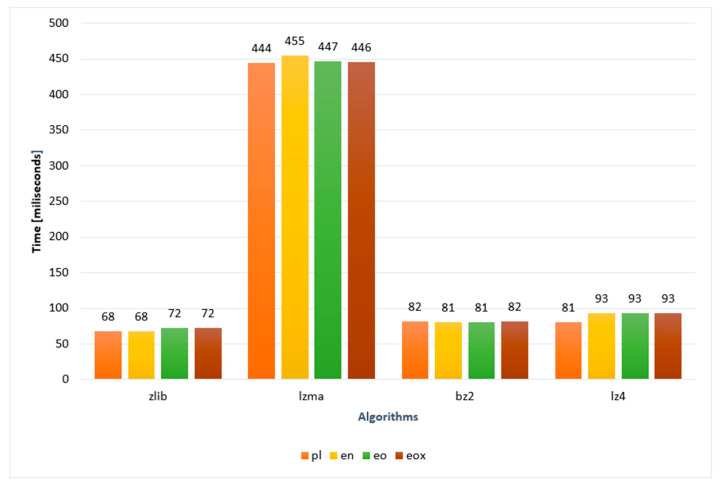
The compression time.

**Figure 8 sensors-22-06393-f008:**
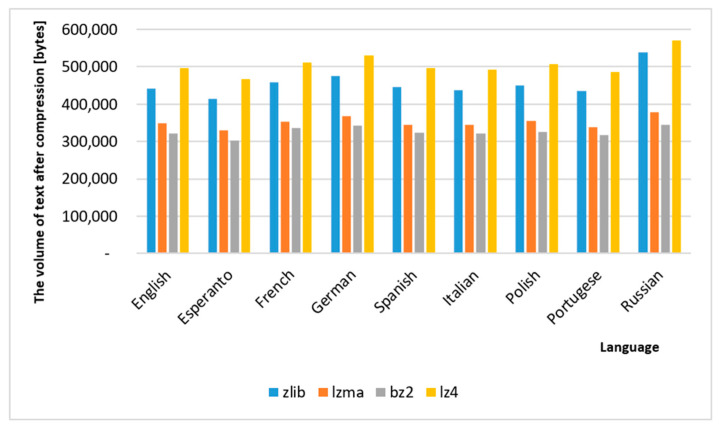
The result of additional experiment—comparison of compression of text translated in Google Translate.

**Figure 9 sensors-22-06393-f009:**
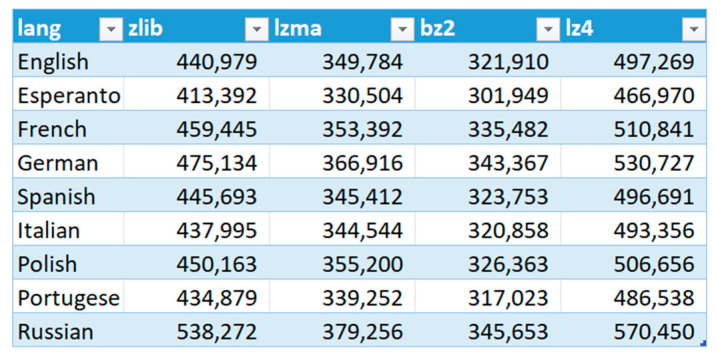
The summarization of the additional experiment (bytes).

**Table 1 sensors-22-06393-t001:** Data on uncompressed text.

Parameter	pl	en	eo	eox
The volume of the uncompressed text (bytes).	1,187,923	1,198,403	1,174,480	1,174,480

**Table 2 sensors-22-06393-t002:** The sample text from *Quo vadis* in different languages and the adequate number of characters.

Quotation	Number of Characters
“I tak minął Nero, jak mija wicher, burza, pożar, wojna lub mór, a bazylika Piotra panuje dotąd z wyżyn watykańskich miastu i światu”.	131
“Therefore, Nero passed, as a whirlwind, as a storm, as a fire, as war or death passes; but the basilica of Peter rules till now, from the Vatican heights, the city, and the world”.	174
“Tiel pasis Nero, kiel pasas uragano, fulmotondro, brulo, milito aŭ pesto, dum la baziliko de Petro regas ĝis nun de la Vatikana altaĵo la urbon kaj la mondon”.	157
“Tiel pasis Nero, kiel pasas uragano, fulmotondro, brulo, milito aux pesto, dum la baziliko de Petro regas gxis nun de la Vatikana altajxo la urbon kaj la mondon”.	160

**Table 3 sensors-22-06393-t003:** Data contained in the console.

Algorithm	pl	en	eo	eox
Compression time [s]				
zlib	0.0683	0.0683	0.0722	0.0723
lzma	0.4449	0.4552	0.4473	0.4464
bz2	0.0821	0.0813	0.0813	0.0821
lz4	0.0818	0.0934	0.0927	0.0927
Space used [%]				
zlib	38.38	37.20	35.80	35.71
lzma	30.66	29.45	28.58	28.56
bz2	27.93	26.86	26.03	25.98
lz4	43.47	41.87	40.52	40.48
Space used [bytes]				
zlib	455,947	445,863	420,472	419,452
lzma	364,232	352,948	335,724	335,400
bz2	331,739	321,942	305,694	305,111
lz4	516,426	501,722	475,912	475,389

**Table 4 sensors-22-06393-t004:** Data on compression efficiency [%].

Algorithm	pl	en	eo	eox
zlib	38.38	37.2	35.8	35.71
lzma	30.66	29.45	28.58	28.56
bz2	27.93	26.86	26.03	25.98
lz4	43.47	41.87	40.52	40.48

**Table 5 sensors-22-06393-t005:** Data on the efficiency of compression in relation to the volume of text in Polish [%].

Algorithm	pl	en	eo	eox
zlib	38.38	37.53	35.40	35.31
lzma	30.66	29.71	28.26	28.23
bz2	27.93	27.10	25.73	25.68
lz4	43.47	42.24	40.06	40.02

**Table 6 sensors-22-06393-t006:** Data on the compression time [ms].

Algorithm	pl	en	eo	eox
Zlib	68	68	72	72
Lzma	444	455	447	446
bz2	82	81	81	82
lz4	81	93	93	93

## Data Availability

Not applicable.

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
