# Peer review of "Compression of Text in Selected Languages—Efficiency, Volume, and Time Comparison"

_sensors, 2022, doi:10.3390/s22176393_

Round 1

Reviewer 1 Report

The presented article concerns the possibility of using the planned language Esperanto for text compression and comparing the results of the text compression in Esperanto with the compression in natural languages, represented by Polish and English.

In general, the paper is interesting and prepared with the necessary care and has a scientific value.

The results of the studies are presented. The authors developed a program for text comparison that can effectively compress text with the use of four algorithms,  and also present the collected information. All four used algorithms indicated that the planned language Esperanto in any form of writing allows for a higher degree of compression than the studied natural languages. 

 I can recommend the paper for possible publication.

Author Response

Thank you for your review. 

Reviewer 2 Report

The authors study the possibility of using the planned language Esperanto for text compression and compare the results of the text compression in Esperanto with the compression in natural languages, represented by Polish and English. The authors performed text compression in the created program in Python using four compression algorithms: zlib, lzma, bz2, and zl4 in four versions of the text: in Polish, English, Esperanto, and Esperanto in x notation (without characters outside ASCII encoding). After creating the compression program, and compressing the proper texts, the authors conducted an analysis on the comparison of compression time and the volume of the text before and after compression. I have some comments:

1- Please underscore the scientific value-added of your paper in your abstract and introduction.

2- The contribution is not clear in the introduction part. The introduction should be clearly stated the research questions and targets first. Then answer several questions: Why is the topic important (or why do you study it)? What are the research questions? What has been studied? What are your contributions? Why is it to propose this particular method? 

3- The authors should strengthen their argument with regard to the scientific novelty and also justify the need to develop this scheme in this paper.

4- The related work section is not well written and many related recent references are required; the authors should insert the results of the previous related works and make a critical analysis by introducing the weaknesses or shortcomings of these works. For example, some recent related data reduction methods (lossy or lossless) can be considered like:

- New fog computing enabled lossless EEG data compression scheme in IoT networks

Authors should add further recent related references to the related work section.

5- The method proposed by the authors should be compared with the existing methods and explained to highlight the advantages of the new method.

6- Authors need to check the writing all over the paper. The paper needs proofreading and improving the presentation.

7- The conclusion should be improved.

Author Response

Thank you for your review. The answers to the comments are in the attached file. The corrections are seen in the new version of our manuscript. 

Reviewer 3 Report

Authors use the compression advantages of simplicity, regularity and repetitive fragments in artificial language Esperanto  to compare the text compression performance with  other languages of Polish and English. Their experiments are conducted on their python programs using existing compression algorithms.

According to the Journal scope and readership, authors are strongly suggested to described how to apply their "text-compression" work contributed to applications related to IoT or intelligent sensors to substantially save resources. I believe authors may extensively spend more paragraphs for some applications such as  "speech to text" or "text to speech" in Introduction, Section 2 or Conclusions. 

In Conclusions, since authors claim they developed a program for text comparison that can "effectively" compress text with the use of various algorithms, authors can give a flowchart or pseudo code for their developed program in Section 2.4. 

BTW, since "Google translation" has supported Esperanto, I suggest that authors can simply demonstrated that Esperanto text compression ratios compared with other natural languages using Google translation, by some input samples such as short paragraphs in Polish or English. Authors may use the same codebook applied in experiments. 

Author Response

(The authors gave the same response as above.)

Reviewer 4 Report

In the present manuscript, the authors study the possibility of using the planned language Esperanto for text compression and compare the results of the text compression in Esperanto with the compression in natural languages. However, I will comment on some aspects to improve the quality of the manuscript and for the authors to highlight the suggested changes:

- The authors have not written the meanings of some acronyms.

- The authors have not written in the correct verb tense according to the section.

- The authors have not written which version they have the software they are using for the proposal.

-The authors do not have a Related Works Section.

The authors must eliminate “[authors’ own study]” because they have performed the research, and it does not require self-reference.

- Figure 2 is not a Figure, although the data can be in a Table or can be shown in a graph.

- The word "formula" must not be mentioned when citing an Equation.

- The authors have not explained why the lzma algorithm spends much time on compression compared to the other algorithms.

- How much is the length or number of characters that you have served for experimentation?

- The authors do not demonstrate what should be entered and what is obtained after the text compression process.

- Are the metrics the authors also evaluated the same when decompressing a text?

- What would be the measurements in the Romance languages? Or in more complicated languages ​​in their writing?

- The quality of the conclusions and future work should be improved.

- The references must not be very old with a maximum of up to 7 years can be considered, and must be improved.

Author Response

(The authors gave the same response as above.)

Round 2

Reviewer 3 Report

Authors have properly revised the manuscript according to the review report.

#

Reviewer 4 Report

The present manuscript cannot be reviewed, due to the poor presentation of the article for a high ranking journal, since there is a gray box on the right side, with strikethroughs that cause confusion to the reviewer. The authors must present the highlighted modifications correctly.